# Food Waste and Nutrition Quality in the Context of Public Health: A Scoping Review

**DOI:** 10.3390/ijerph18105379

**Published:** 2021-05-18

**Authors:** Aoife Brennan, Sarah Browne

**Affiliations:** 1School of Public Health, Physiotherapy & Sports Science, University College Dublin, Dublin 4, Ireland; aoife.brennan4@ucdconnect.ie; 2Institute of Food & Health, University College Dublin, Dublin 4, Ireland

**Keywords:** food waste, food surplus, nutrition, food security, public health nutrition, healthy eating, sustainability, environmental health

## Abstract

Food waste and nutrition are intrinsically linked in terms of environmental health and public health. Despite this, it is unknown whether these topics have been previously synthesized into a review. The aim was to identify the interdisciplinary parameters that exist in public health and nutrition literature in terms of food waste and plastic waste associated with food, and to identify how these parameters currently contribute to food sustainability messaging and interventions. A rapid scoping review was conducted. Data were mapped into concepts and synthesized in a narrative review. Four main concepts were identified: (1) food waste and diet quality, nutrient losses, and environmental health, (2) food waste reduction interventions and diet quality, (3) food banks/pantries and diet/nutritional quality, and (4) food and plastic waste messaging in nutrition or dietary guidelines. Food waste is associated with nutrient wastage, and interventions to reduce food waste can successfully address food sustainability and nutrition quality. Food redistribution systems do not currently address access to sustainably sourced foods that are also nutrient-dense for lower-income communities. Opportunities for future research and practice include aligning food waste, plastic waste, and nutrition priorities together and developing better food redistribution systems to limit wastage of high-quality foods.

## 1. Introduction

Globally, the implementation of sustainable practices has become a key priority, which includes a shift toward more sustainable food systems [1]. A sustainable food system is defined as “a food system that delivers food security and nutrition for all in such a way that the economic, social, and environmental bases to generate food security and nutrition for future generations are not compromised” [2]. Our current global food systems are not in line with this definition and cannot be considered sustainable, as they do not provide food security and have numerous negative environmental impacts [3,4,5]. Currently, reductions in food waste and plastic waste are important strategies in the move toward more sustainable consumption and production patterns [1]. As part of the Sustainable Development Goals, the United Nations aims to “halve per capita global food waste at the retail and consumer levels” and to “substantially reduce waste generation through prevention, reduction, recycling, and reuse” by 2030 [1]. Policies, incentives, and campaigns to reduce food and plastic waste at production and consumer stages also have the potential to address public health by identifying simultaneous opportunities for nutrition and healthy eating messaging.

Food loss and food waste occur at varying stages of the food supply chain (Figure 1). The food supply chain describes how food products pass from producer to consumer; this includes the stages of production, processing, distribution, retail, and consumption. Food loss refers to the loss of food along the stages of the food supply chain that lead to edible food for human consumption [6,7]. Food waste refers to losses that occur at the end of the food supply chain, specifically at the retail and consumption stages, and is linked to consumer and retailer behavior [6,7]. Food loss and waste and public health intersect in the domains of food security and nutrition [8]. In 2019, two billion people (25.9% of the global population) were affected by hunger or did not have access to sufficient, nutritious food [5]. On the other hand, approximately one-third of edible food (1.3 billion metric tons) produced for human consumption is lost or wasted along the food supply chain each year [6]. There are also significant nutritional losses embedded in food waste, and food with a higher nutritional profile, such as fruit and vegetables, tend to be the most wasted [7,9,10,11]. As such, the reduction or redistribution of food waste has the potential to increase access to food, while improving nutrition and diet quality. From an environmental health perspective, food loss or waste represents a waste of the resources involved in production, including land, water, fertilizer, pesticides, and energy, while also contributing to unnecessary carbon dioxide (CO_2_) emissions [12,13,14,15,16,17,18,19]. Food waste also places an unnecessary burden on waste management systems, with varying environmental impacts depending on the system utilized [20,21].

Current food waste reduction strategies based on a food waste hierarchy include reducing the amount of food waste generated (prevention), redistribution of surplus food, the use of food waste in animal feed and industry, composting, anaerobic digestion, and disposal [22]. Effective strategies to reduce waste at the consumer level include consumer awareness campaigns [23,24,25], nudge interventions, such as reduced portion sizes or the implementation of trayless dining [26,27,28], and retail initiatives, such as the Ugly Fruit Campaign which promotes the sale of imperfect fruit and vegetables to reduce food waste [25,29,30]. Food waste reduction strategies at a household level include meal planning, effective use of leftovers, correct storage of food, and the avoidance of overconsumption and excess purchasing. However, while these prevention strategies are effective at reducing waste, it is unclear whether they have a direct effect on diet quality and nutrition. Redistribution of surplus food typically centers around food recovery programs, food banks, and food pantries, whereby surplus food that would otherwise be diverted to landfill or other waste systems is distributed to those in need [31,32,33,34,35]. Food surplus is food produced beyond our nutritional needs, and a large proportion of food loss or waste at each stage of the food system arises from surpluses in supply [22]. As such, effective and well-developed redistribution systems can address waste, while improving food security in vulnerable populations.

Food packaging systems are an integral part of building more sustainable food systems. Plastic is one of the most conventional materials used in food packaging and plays an important role in minimizing food loss and waste by extending the shelf-life of food and protecting food from chemical and biological contamination and physical damage [36]. However, plastic packaging has numerous environmental impacts and the extent of plastic packaging waste currently generated is unsustainable [37,38]. Approximately 25.8 million metric tons of plastic waste is generated in Europe every year, a large proportion of which is attributed to plastic packaging [38]. Only 30% of this plastic waste is returned for recycling, while 31% and 39% are sent to landfill or for incineration, respectively [38]. This allows large quantities of plastic waste to leak into the environment, while incineration releases large quantities of CO_2_ [38].

Food packaging is also linked with nutrition and public health. Packaged food and beverages are predominantly categorized as ultra-processed or highly processed foods on the basis of the NOVA and/or Poti et al. classification systems, and they are generally defined as unhealthy, with less favorable nutrient profiles than less processed foods [39,40,41,42,43]. The consumption of these packaged ultra-processed foods has been linked to poorer diet quality and numerous adverse health outcomes [43,44,45,46,47,48]. There is also the potential for chemicals to transfer from plastic food packaging into items of food, which can impact human health via endocrine disruptions [49,50]. Lastly, plastic packaging does not degrade; it is fragmented into smaller particles, known as microplastics, over time. These microplastics can enter the food chain and can, ultimately, be consumed by humans [51,52]. The effects of microplastic consumption are not fully understood but a range of potential physical and chemical effects are being investigated [51].

Despite clear links among food waste, plastic waste, and nutrition in the context of environmental health and public health, it is currently unknown whether these topics have been previously synthesized into a review. The aim of this scoping rapid review was to identify the interdisciplinary parameters that exist in the public health, nutrition, and environmental health literature in terms of food waste and plastic waste, and to identify how these parameters currently contribute to public health and food sustainability messaging and interventions.

## 2. Materials and Methods

A rapid scoping review was conducted using the methodological framework devised by Arksey and O’Malley, and refined by Levac et al., as a guide [53,54]. The definition adopted here was to map the existing literature to determine the volume and coverage of the topic, ascertain the types of literature available, and identify the gaps in the existing research [54]. A scoping review fit with the aims and objectives of the research question which sits across diverse research disciplines including food systems, public health nutrition, and environmental health. It allowed for a broad search that incorporated a range of publication types from both published and gray literature, where parameters in the context of environmental health and public health could be conceptualized and mapped for potential synergies. The review was reported following the PRISMA Scoping Reviews (PRISMA-ScR) Checklist [55].

### 2.1. Protocol

The research question, inclusion criteria, and search strategy were predefined in a protocol, which was developed by two researchers (A.B. and S.B.).

### 2.2. Eligibility Criteria

Publications concerning food waste and/or plastic waste in combination with nutrition and/or diet quality analysis in the context of environmental health and public health were included. All study types were eligible for inclusion. The outcomes of interest included nutrition (measures of nutrition quality, diet quality, and nutrient losses), environmental impacts (measures of land, water, pesticide, fertilizer, CO_2_, and ecological losses), food waste (measures of the amount and types of food waste generated), food safety (presence of plastics and chemicals in food), or plastic waste (measures of quantity/generation of plastic waste). Another area for inclusion was studies discussing the nutritional quality of food distributed by food redistribution systems, such as food banks and food pantries. These redistribution systems have been shown to effectively redistribute food surpluses, to be a key component of the food waste hierarchy [22], and to have the potential to impact public health. These studies were eligible for inclusion without direct measures of food waste, as research has shown that a large proportion of food distributed from these centers is from donations of food surpluses/waste [31,32], and it was inferred that this was applicable to all food redistribution systems. Interventions, policies, directives, strategies, and guidelines that discussed food waste and/or plastic waste, as well as nutrition, in the context of environmental health and public health were also eligible for inclusion.

### 2.3. Information Sources and Literature Search

A formalized search of three databases was conducted between January and February 2021. Relevant publications were identified via a computerized search of Scopus, MEDLINE, and Global Health using relevant search terms and medical subject headings (MESH). This included variations and combinations of plastic waste and food waste, research, policy and practice, public health, nutrition, environmental health, environmental nutrition, food environment, food sustainability, food security, and food safety. Scopus and MEDLINE were selected as they would provide a comprehensive overview of the literature available in this area, which is fitting with the purpose of a scoping review. Global Health was selected as it would allow for the identification of literature in this area specifically in the context of public health, which was one of the key outcomes of this scoping review. The full search strategy is available in Appendix A. Additional publications were identified via handsearching the reference lists of relevant publications and gray literature, including Google Scholar, publications by the European Commission, Government publications, and publications by relevant stakeholder organizations, e.g., the Waste and Resources Action Program (WRAP). The search was limited to publications in the English language published between 2010 and 2021. The aim of the review was to identify recent literature and determine the gaps in current research. Research in this area has increased rapidly within this timeframe, with a more urgent focus on changes needed to develop a sustainable food system. The search strategy was developed by two researchers (A.B. and S.B.) and conducted independently by one researcher (A.B.).

### 2.4. Selection of Sources of Evidence

Screening was conducted in three phases and all publications were screened against the eligibility criteria. Phase one involved screening the title and abstract of relevant publications. This was conducted by one researcher (A.B.), with a second researcher screening 10% of these publications (S.B.). The researchers met between phase one and phase two to discuss agreement and resolve any conflict in screening. Full-text articles were screened in phase two by one researcher (A.B.). Phase three of screening involved mapping the concepts for inclusion. During phase three, several potential concepts were identified. Three of these concepts (‘food contact chemicals, plastic waste, and public health’, ‘food packaging advances’, and ‘microplastics and seafood’) were excluded as they were lacking the nutrition/diet quality element of the inclusion criteria, while another concept (‘ultra-processed/packaged foods and nutrition/diet quality’) lacked the food waste or plastic waste elements of the inclusion criteria. After screening, two researchers (A.B. and S.B.) met to discuss included and excluded publications and agree a consensus on the final included publications and concept categories.

### 2.5. Data Charting Processes and Data Items

Relevant data items were abstracted from each of the selected publications, including the title, author, and year of publication, the publication type, and the results and/or key messages in relation to food waste, plastic waste, and nutrition in the context of environmental health and public health. Data abstraction was conducted by one researcher (A.B.).

### 2.6. Critical Appraisal of Individual Sources of Evidence

The selected publications were not critically appraised, which is in line with current scoping review guidelines [56,57].

### 2.7. Synthesis of Results

The results from the data abstraction process were described and a narrative synthesis presented. Outcome measures were mapped into several concepts for the basis of discussion. The key areas identified included (1) food waste and diet quality, nutrient losses, and environmental health, (2) food waste reduction interventions and diet quality, (3) food banks/pantries and diet/nutritional quality, and (4) food waste and plastic waste in nutrition or diet guidelines.

## 3. Results

The search yielded a total of 6649 results. The screening, selection, and exclusion processes of these results are depicted in Figure 2. A total of 33 publications were eligible for inclusion in the final review.

### 3.1. Characteristics of Sources of Evidence

A range of publication types were included in this review, including 10 cross-sectional studies [58,59,60,61,62,63,64,65,66,67], one review [68], two systematic reviews [69,70], two case studies [71,72], seven intervention studies (three non-randomized controlled trials [73,74,75], one randomized controlled trial [76], three pre-post-design studies [77,78,79]), one observational study [80], one comparative analysis [81], one critical evaluation [82], one time series [83], one multi-method study [84], and six food-based dietary guidelines [85,86,87,88,89,90]. Four main concepts were mapped from the resulting data (Figure 3), and the publications were grouped into these concepts as appropriate. Characteristics of included studies and reported outcomes are summarized for each concept in Table 1, Table 2, Table 3, Table 4 and Table 5.

### 3.2. Synthesis of Results by Concept

#### 3.2.1. Concept 1: Food Waste, Diet Quality, Nutrient Losses, and Environmental Health

The characteristics of studies in Concept 1 are summarized in Table 1. Food waste measurements ranged from 107 g–422 g per capita per day, 65–110 kg per capita per year, [59,61,62,64], and 2.98 kg per week [80]. Six studies analyzed food waste at consumer level [58,59,61,62,64,80] and two studies analyzed food waste at retail and consumer levels [60,63]. Six studies assessed the nutritional losses embedded in wasted food, with quantities of wasted nutrients varying significantly between studies and different nutrients being identified as the most wasted [58,61,62,63,64,80]. Five of these studies assessed losses in terms of per capita per day which are summarized in Table 2. Six studies assessed the link between environmental health and food waste, with loss of land, water, pesticides, fertilizers, and unnecessary CO_2_ emissions being the most cited environmental impacts [60,61,62,64,80]. Fruit and/or vegetables were consistently ranked the most wasted foods [58,59,61,62,63,64,80].

The link between diet quality and food waste was assessed in three studies [59,64,68]. Three studies assessed diet quality using the Healthy Eating Index 2015 (HEI-2015) [59,60,64], with one of these studies also using the Alternative Health Eating Index 2010 (AHEI-2010) as a comparison [60]. Higher diet quality was associated with greater amounts of overall food waste in two studies [60,64], and greater amounts of fruit and vegetable waste only in the other [59]. The link between diet quality and environmental health was discussed in two studies [60,64]. Higher diet quality was associated with less land use, but greater waste of irrigation water and pesticides depending on the measure used to assess diet quality [60].

#### 3.2.2. Concept 2: Current Interventions Aimed at Preventing/Reducing Food Waste, While Improving Diet Quality and/or Nutrition

The characteristics of studies included in Concept 2 are summarized in Table 3. Six interventions were school-based interventions that aimed to reduce food waste while improving nutrition knowledge, diet quality, and/or nutritional intake [73,74,75,76,77,78]. Three of four education intervention studies reported decreases in food waste and improvements in nutritional intake in the intervention groups in comparison to the control groups, including maintaining or increasing fruit and vegetable intake or increasing the consumption of nutritionally balanced school meals [73,74,75]. One study did not find any improvement in dietary intake or plate waste [76]. Two school-based interventions implemented optimized menus, which were not successful at reducing food waste but demonstrated a decrease in calculated greenhouse gas emissions (GHGE) while maintaining nutritional adequacy [77,78]. Another intervention examined the effect of implementing reduced portion sizes on plate waste in two dining facilities [79]. Plate waste was reduced at both sites, and intakes of energy, fat, saturated fat, cholesterol, sodium, fiber, calcium, potassium, and iron were also reduced [79].

#### 3.2.3. Concept 3: Food Banks/Pantries and Diet/Nutritional Quality

The characteristics of studies included in Concept 3 are summarized in Table 4. Eleven studies assessed the link among food banks/pantries, nutrition, and diet quality [65,66,67,69,70,71,72,81,82,83]. In the studies that analyzed the nutritional content of the food being distributed by food banks/pantries, a range of nutritional issues were identified and are presented in Figure 4 [65,66,67,69,70,81,82,83]. Three case studies demonstrated how effective planning and redistribution can increase the nutritional quality of the food being distributed by food recovery programs, food banks, or food pantries, while reducing food waste [71,72,84] and environmental impacts [84].

#### 3.2.4. Concept 4: Food Waste and Plastic Waste in Nutrition or Diet Guidelines and Policies

The key messages in guidelines or policies that were included in Concept 4 are summarized in Table 5. Six food-based dietary guidelines (FBDGs) incorporated consumer information on the reduction of food waste [85,86,87,88,89,90]. Two FBDGs also addressed reducing food packaging waste in conjunction with food waste reduction strategies [87,89]. The most common messages in relation to reducing food waste are presented in Figure 5. Overall, food and/or plastic waste messaging was consumer-orientated.

## 4. Discussion

The purpose of this rapid scoping review was to identify peer-reviewed literature, gray literature, and policy documents exploring the existing research, practice, and policy that incorporate the relationship among plastic waste and/or food waste and nutrition in public health and environmental health contexts. Four key concepts that pertain to nutrition and/or diet quality in combination with food waste were identified.

The first concept identified was the link among food waste, wasted nutrients, diet quality, and environmental health. Notably, the volume of food wasted per capita and most wasted nutrients varied considerably across studies [58,59,60,61,62,63,64]. This may be due to several reasons, including the country/countries the study was conducted in, their dietary practices and associated waste practices, the sample size of participants in the study, and the methods used to measure food waste [91]. Variation in results is not uncommon across food waste studies [92].

The environmental impacts associated with food waste included loss of land, water, pesticides, and fertilizers, as well as unnecessary GHGE. These results are similar to findings from other studies [15,17,19]. Overall, fruit and vegetables were identified as the one of the most wasted food groups when food waste was compared by weight. This is in line with several other studies [10,11,93] and is an important consideration as fruit and vegetable losses have been shown to result in some of the greatest losses of nutrients, such as fiber and carotenoids [62,63,80]. The production of wasted fruit and vegetables has also been shown to account for some of the greatest losses of irrigation water, crop land, and pesticides [64], and it is responsible for a large proportion of CO_2_ emissions [80]. As such, interventions that promote increased fruit and vegetable consumption, alongside a reduction in food waste, could have the greatest benefits in terms of public health nutrition while also reducing environmental impacts, as studies have shown that increasing fruit and vegetable consumption is effective at improving nutritional biomarkers, such as vitamin C, folate, and carotenoids [94].

There is also a link among diet quality, food waste, and environmental health. The HEI-2015 was used to calculate diet quality in the three studies that discussed this link [59,60,64], with one study using the AHEI-2010 as a second measure for comparison [60]. The HEI-2015 assesses whether intakes are in line with the Dietary Guidelines for Americans (DGA) based on 13 components, with a higher score being indicative of higher diet quality [95]. Based on this measure, higher diet quality is associated with higher intakes of fruit, vegetables, wholegrains, dairy, protein foods, and fats. Overall, higher diet quality was associated with greater amounts of food waste in two studies [60,64] and greater amounts of fruit and vegetable waste only in the third study [59]. This link between increased food waste and diet quality may be explained by the high wastage of food groups associated with higher diet quality as identified in this review and in previous research [10,11,58,59,61]. However, there are likely additional factors driving this association, including household income (there may be less waste in lower-income households), household size and composition (households with children are likely to waste more), and household demographics and culture [7]. In terms of environmental losses, greater HEI-2015 scores were associated with less land use but greater use of irrigation water and pesticides, while higher AHEI-2010 scores were associated with less land use and a similar use of irrigation water and pesticides [60,64]. This disparity in results could be due to the differences in the scoring standards between the two measures, particularly in the categories of fruit, meat, and dairy.

Our second objective was to report and characterize food sustainability messaging and interventions in research and public health contexts. Several interventions that aimed to reduce food waste, while improving nutrition or diet quality, were identified [73,74,75,76,77,78]. School-based interventions with a nutrition education component have been shown to reduce food waste while improving nutritional intake, including increasing fruit and vegetable consumption [73,74,75]. Reducing food waste while improving dietary quality in school settings is an area that is beginning to receive greater attention. For example, the EIT Food School Network Program, which was established in 2018, is working toward improving dietary intake while reducing food wastage in schools across Europe [96]. The implementation of reduced portion sizes in dining facilities has also proven to be effective at reducing plate waste, while reducing intakes of energy, cholesterol, fat and saturated fat, calcium, and iron [79]. The association between reduced portion sizes and reduced plate waste has been identified in several other studies, but not in the context of the effects on nutritional intake [27,28]. Reduced portion sizes have the benefit of reducing plate waste, while simultaneously preventing energy overconsumption, which is linked to the incidence of overweight and obesity [97]. However, improving nutritional intake is complex and important food components like fiber, vitamins, and minerals such as iron and calcium need careful monitoring to maintain dietary quality while consuming less.

Food banks and pantries are effective means of redistributing surplus food wherever they arise in the food system and can simultaneously improve access to food and reduce the negative environmental impacts associated with food waste. However, our results indicate that food banks are not meeting the nutritional needs of users, and several nutritional issues have been identified with the packages distributed from these centers (Figure 4) [65,66,69,70,81,82,83]. One of the most common issues was insufficient micronutrient provision. Given that fruit and vegetables have been identified as the most wasted food group and that packages are low in the nutrients typically found in these foods, the redistribution of more fruit and vegetable surpluses to food banks could be an effective way to address both the issues of food waste and inadequate micronutrient provision in food packages. Furthermore, fruit and vegetable intakes are generally low in the cohort that utilizes food banks [98,99], making the provision of these foods of particular importance, as users are already at risk of micronutrient deficiencies [99]. Several issues have been identified with increasing fruit and vegetable provision in food banks, including variations in produce availability, transportation times, and inadequate storage facilities [100]. However, successful implementation is possible, as evidenced by the Leket food bank in Israel. This food recovery program has implemented the large-scale rescue and redistribution of perishable foods, while focusing on nutritional quality [71]. The fruit and vegetable portions contributed by Leket Israel have been shown to correlate positively with the dietary quality of food packages [67].

Food bank users are also more likely to have poorer health and suffer with overweight/obesity, diabetes, and hyperlipidemia [101,102]. As such, it is concerning that food bank packages have been shown to provide excess energy, carbohydrates, and fat, as excessive consumption of these nutrients is linked to these nutrition-related chronic diseases. Current strategies employed by food banks to improve the nutritional quality of food packages and the diet quality of food bank users include building a healthier inventory, enhancing the access, distribution, and storage capacity of partner programs, community-based nutrition/culinary skills education, and expanding community partnerships/intervention settings for healthy food distribution [103,104]. Overall, improving the nutritional quality of the food provided by food banks and pantries is important and, until they meet the nutritional requirements of their users, it cannot be said that they are fully addressing the issue of food insecurity. Access to healthy food may be another way to combat health inequalities and improve the ability of all sectors of society to participate in a sustainable food system. However, food poverty is a systemic issue, and improving access to nutritious food is only one step toward eradicating health inequalities and the move toward more sustainable food practices. A more long-term and holistic approach to improving dietary behaviors and increasing food security, alongside short-term solutions provided by food pantries, includes addressing systemic and social reasons for food, hunger, and poverty, as well as reducing health inequalities.

In terms of sustainability messaging, several FBDG discussed the reduction of food waste [85,86,87,88,89,90]. Two mentioned the reduction of plastic packaging, but recommendations were minimal [87,89]. FBDGs were the only instances found where food waste and/or plastic waste and nutrition were discussed in combination within research, policy, and practice (as identified during this review process). Future dietary guidelines should aim to integrate sustainability and healthy eating messaging, including recommendations to minimize food waste and plastic waste while improving nutrition, as this provides an opportunity to simultaneously improve public and environmental health. Overall, the Qatar FBDGs were the most comprehensive in terms of sustainability and nutrition messaging [87]. These guidelines provide healthy eating advice alongside actionable measures to reduce food and plastic waste and, as such, could be used as a starting point for countries aiming to improve and update their own FBDGs. Going forward, it may be beneficial to include more specific advice around the reduction of food waste in FBDGs. Current messages, while important, are quite general (Figure 5). As such, it would be useful to provide more specific advice, perhaps pertaining to the most wasted food groups such as fruit and vegetables, including preparation, storage, and leftover recommendations.

Additional policies, strategies, and directives that discussed food waste and plastic waste were identified but lacked the nutritional aspect of the eligibility criteria [38,105,106,107,108]. Food waste and plastic waste were typically discussed together in the context of a circular economy [105,108]. Several additional areas were identified where plastic waste overlapped with public health, including microplastics in the food chain and food contact chemicals; however, again, the nutrition element of the eligibility criteria was lacking. Overall, this scoping review identified that plastic waste, nutrition, and public health are rarely discussed in combination, and this highlights the need for future research, as well as the development of policy and practice, in this area.

Considering the current COVID-19 pandemic, it is of particular importance that links between plastic waste, nutrition, and public health are investigated. COVID-19 has led to a change in our waste generation patterns and a surge in the demand and use of plastics, which has had subsequent effects on our waste management systems [109]. There is a perception that plastic is more hygienic, which has led to a shift in consumer choice in favor of plastic packaging and single-use plastic bags [110]; furthermore, reusable bags have been banned in certain areas [110], lockdowns and home quarantines have led to an increase in online shopping and home delivery of food and groceries which may increase the generation of packaging waste [111], and the use of disposable utensils has increased due to convenience and safety [110]. As such, it is essential that we understand the implications that this may have for public health.

### 4.1. Implications for Practice

Overall, food waste, plastic waste, and nutrition are rarely discussed in combination despite being inherently linked and key components in the transition toward more sustainable food systems. There are opportunities in future research, policy, and practice to address these issues together to align public health and environmental health priorities. Institutional interventions (education, childcare, healthcare, residential care, and prison systems for example) that simultaneously address sustainable food sourcing, nutrition quality, and food waste are potential high-impact initiatives at national levels that could be prioritized to progress many key national targets for public health and environmental sustainability. Addressing the nutrition quality of food banks at larger scales and developing nutrition and sustainability policies for this sector would improve access to sustainable diets for low-income and at-risk groups. Lastly, FBDGs are regularly reviewed and there are opportunities to learn from countries leading the way in terms of incorporating food and plastic waste messaging alongside healthy eating. Beyond the studies included in the current review, there are several additional areas in relation to food waste, plastic waste, and nutrition that could be explored further, including microplastics and the food chain, ultra-processed foods, and food contact chemicals.

### 4.2. Strengths and Limitations

This rapid scoping review was conducted in line with a well-established framework and reported according to the PRISMA-ScR checklist. While the researchers aimed to be as comprehensive as possible with the search, all relevant publications in the published and gray literature may not have been identified. Only publications in the English language and those published between 2010 and 2021 were included. Furthermore, data abstraction was conducted by one researcher and, as such, the characterization and interpretation of the results may have been subject to bias.

## 5. Conclusions

In conclusion, this review identified links among food waste, nutritional losses, diet quality, and environmental health. Examples from educational settings demonstrate that sustainable food sourcing can be integrated with healthy eating and food waste reduction. Food redistribution systems may be effective at redistributing surplus food and reducing waste but do not meet the nutritional needs of their users. As such, they are not adequately addressing food insecurity for low income and ‘at-risk’ groups, and interventions to improve nutritional quality are required. FBDGs are beginning to address food waste, plastic waste, and nutrition together in consumer-orientated messages. While public health messages are important for consumer awareness, supportive interdepartmental policies and greater interdisciplinary research are needed to address macrolevel systems that can improve access for all to affordable sustainable food systems.

## Figures and Tables

**Figure 1 ijerph-18-05379-f001:**
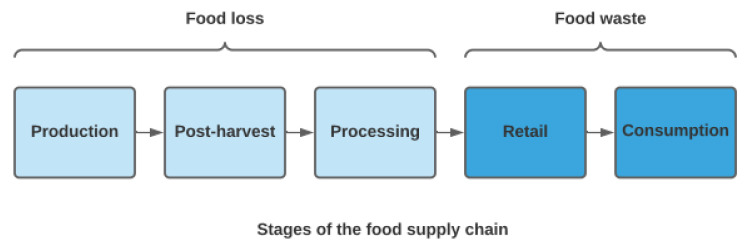
The stages of the food supply chain associated with food loss and food waste [6,7].

**Figure 2 ijerph-18-05379-f002:**
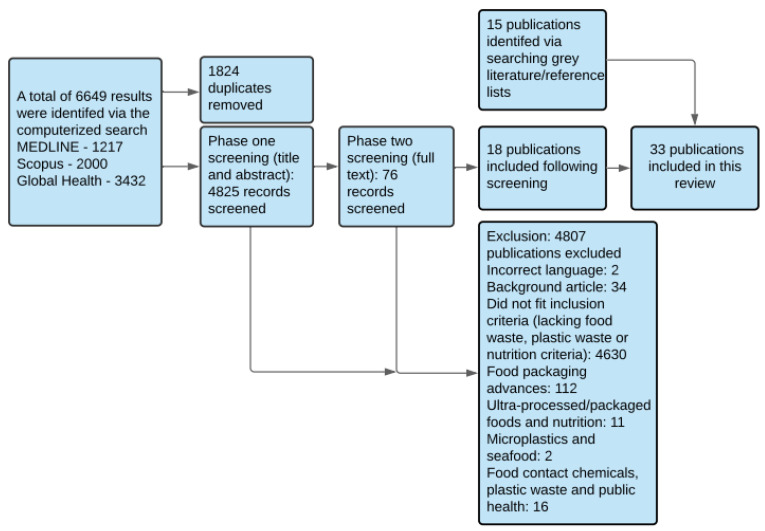
Flow chart of the search, screening, and selection process, including details of included and excluded studies and concepts.

**Figure 3 ijerph-18-05379-f003:**
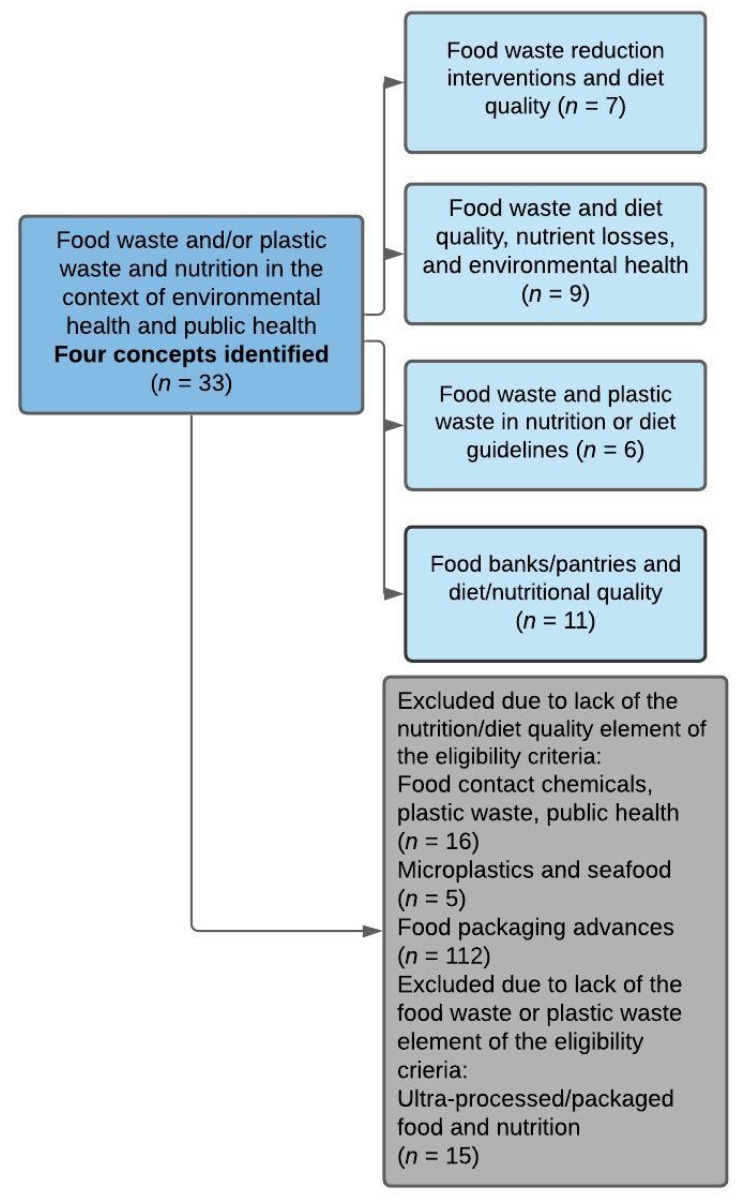
Map of studies and concepts included and excluded, measured by number of studies included in each concept (studies were identified via a computerized search and handsearching gray literature/reference lists). Concepts depicted in gray were excluded following phase three of screening.

**Figure 4 ijerph-18-05379-f004:**
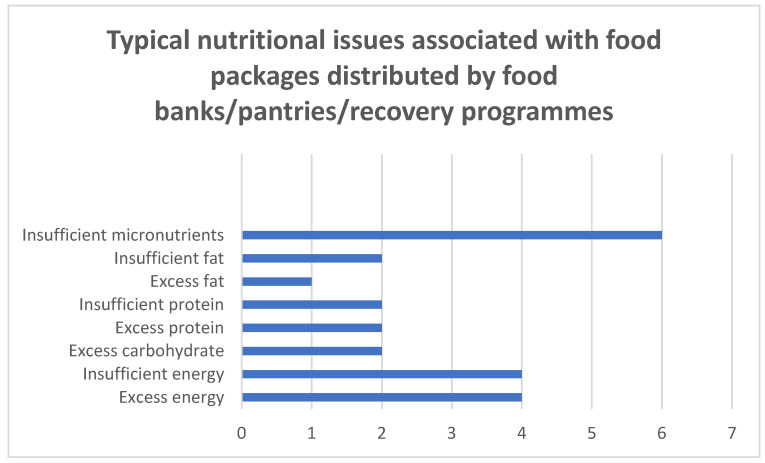
The nutritional issues and the frequency of nutritional issues associated with food packages distributed by food banks/pantries reported by eight studies [65,66,67,69,70,81,82,83].

**Figure 5 ijerph-18-05379-f005:**
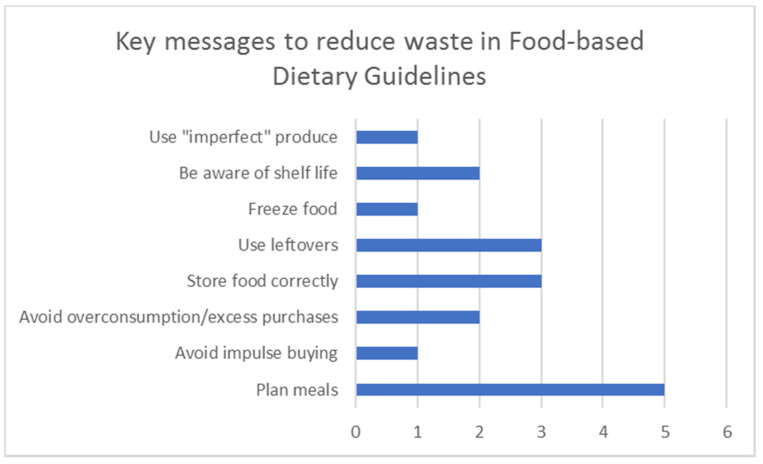
Key messages related to food waste that featured in food-based dietary guidelines included in this review and the frequency in which they appeared [85,86,87,88,89,90].

**Table 1 ijerph-18-05379-t001:** Results of studies included in Concept 1: food waste, diet quality, nutrient losses, and environmental health.

Title, Author, Year	Study Type	Results			
		Food waste	Nutrient Waste	Diet Quality	Environmental Impacts of Food Waste
Assessment of nutritional loss with food waste and factors governing this waste at household level in Pakistan [58]. Khalid et al. (2019)	Exploratory study Cross-sectional	Cooked food, fruit, and vegetables were the most wasted.	Household food waste led to a loss of 54.42 kcal, 2.61 g of protein, 2.21 g of lipids, 10.58 g of carbohydrate, 0.75 g of fiber, 275.2 mcg of beta-carotene, 22.49 mg of calcium, 96.83 RE of vitamin A, and 37.11 mg of phosphorus per capita per day.	N/A	N/A
Association between diet quality and food waste in Canadian families: a cross-sectional study [59]. Carroll et al. (2020)	Cross-sectional study	Households produced an average of 107 g of avoidable and 52 g of unavoidable food waste per capita per day. Fruit and vegetables were the most wasted foods.	N/A	Overall, diet quality was not associated with total daily per capita food waste. Parent diet quality was positively associated with daily avoidable and unavoidable fruit and vegetable waste. Diet quality was assessed using the HEI-2015, with higher scores being indicative of higher diet quality.	N/A
Healthy diets can create environmental trade-offs, depending on how diet quality is measured [60]. Conrad et al. (2020)	Cross-sectional study	Daily per capita total food demand was 1675 g; 7% (111 g) was lost at retail level, 16% (245 g) was inedible, and 31% (410 g) was wasted at consumer level.	N/A	Higher diet quality was associated with greater retail losses, inedible portions, and consumer waste.	One-quarter of agricultural inputs used to produce total food demand were attributed to food that was never consumed. Higher diet quality was associated with lower use of agricultural land. Using the HEI-2015, higher diet quality was associated with greater use of irrigation water and pesticides. This association was not found using the AHEI-2010.
Nutrition in the bin: a nutritional and environmental assessment of food wasted in the UK [61]. Cooper et al. (2018)	Cross-sectional	The total weight of UK household waste was 110 kg per capita per year, of which 77% is thought to be avoidable. Approximately 42 daily diets are disposed of per person each year. Fresh vegetables and salad (25%), drink (13%), bakery (11%), dairy/eggs (8%), complete meals (8%), other foods (8%), meat/fish (7%), and fresh fruit (6%) were the most wasted foods.	The most wasted nutrients were vitamin B12, vitamin C, and thiamine.	N/A	The greenhouse gas emissions associated with wasted edible household food are 0.9 kg CO_2_ equivalents per capita per day or 320 kg CO_2_ equivalents per capita per year. Food waste also contributes to freshwater consumption scarcity, nonrenewable resource depletion, land use, and negative environmental impacts.
Nutritional and environmental losses embedded in global food waste [62]. Chen et al. (2020)	Cross-sectional study	Globally, an average of 65 kg of food waste is generated per capita per year (178 g per capita per day), which accounts for 18 daily healthy diets. The most wasted foods were vegetables (25%), cereals (24%), and fruit (12%). These food groups also contributed to the largest amount of wasted nutrients.	On average, 273 kcal of energy is wasted per capita per day. The most wasted nutrients were vitamin C, K, zinc, copper, manganese, and selenium.	N/A	Wasted food contributes to the loss of 124 g CO_2_ equivalents, 58 liters of freshwater, 0.36 m^2^ of land, 2.90 g of phosphorus, and 0.48 g of nitrogen per capita per day.
Wasted food, wasted nutrients: nutrient loss from wasted food in the United States and comparison to gaps in dietary intake [63]. Spiker et al. (2017)	Cross-sectional study	N/A	Food wasted at retail and consumer levels contained 1217 kcal, 33 g of protein, 5.9 g of dietary fiber, 1.7 μg of vitamin D, 286 mg of calcium, and 880 mg of potassium per capita per day.	N/A	N/A
Relationship between food waste, diet quality, and environmental sustainability [64]. Conrad et al. (2018)	Cross-sectional study	Consumers wasted 422 g of food per person per day. Fruits, vegetables, and mixed fruit and vegetable dishes represented 39% of the total food wasted.	Over 800 kcal was wasted per person per day. The highest micronutrients wasted were carotenoids.	N/A	Annually, wasted food was grown on 7.7% of all harvested cropland in the USA. Over 60% of the land used to grow fruit, 56% of the land used to grow vegetables, and 30% of the land used to grow sweeteners were wasted. Annually, 4.2 trillion gallons of irrigation water, 780 million pounds of pesticides and 1.5 billion pounds of phosphorus fertilizer were used on wasted cropland. Higher diet quality (based on the HEI-2015) was associated with greater food waste, less land use, and greater use of irrigation water and pesticides
Valuing the multiple impacts of household food waste [80]. Von Massow et al. (2019)	Observational study as part of the Family Food Skills study.	An average of 2.98 kg of avoidable waste was generated per household each week. Fruit and vegetables contributed to 66% of total avoidable food waste.	The average household wasted 3366 kcal, 64 g of fiber, 50 mcg of vitamin D, 2 mcg of vitamin B12, 434 mg of vitamin C, 1729 mcg of vitamin A, 1192 mg of calcium, and 675 mg of magnesium per week.	N/A	The global warming potential of avoidable food waste was 23.3 kg of CO_2_ per household per week. Fruit and vegetables represented 40% of the CO_2_ associated with avoidable waste. Avoidable waste was associated with the equivalent of 6.7 m^2^ of land and 5.0 m^3^ of water usage per household per week.
Identifying the links between consumer food waste, nutrition, and environmental sustainability: a narrative review [68]. Conrad and Blackstone (2020)	Review	Discussed definition of food loss/waste, the amount and types of food lost or wasted throughout the food system, the drivers of consumer waste, and reduction strategies.	Discussed links between food waste and wasted nutrients.	Discussed links between diet quality and food waste; higher diet quality is associated with greater amounts of food waste.	Discussed food waste and environmental sustainability; food waste contributes to losses of energy, water, land, pesticides, and fertilizers, and contributes to GHGE.

HEI: Healthy Eating Index, AHEI: Alternative Healthy Eating Index, GHGE: greenhouse gas emissions, CO_2_**:** carbon dioxide, N/A: not applicable to study.

**Table 2 ijerph-18-05379-t002:** Nutrient losses embedded in food waste per capita per day recorded by studies examining the nutrition quality associated with food waste.

Nutrient	Range Wasted Per Capita Per Day Across Studies *
Energy (kcal)	54.4–1216.5 [58,62,63,64]
Protein (g)	2.61–32.8 [58,62,63]
Lipids (g)	2.21–57.2 [58,63]
Carbohydrate (g)	10.58–146.4 [58,63]
Fiber (g)	0.75–5.9 [58,61,62,63]
Vitamin A (ug)	88–308.1 [58,62,63]
Vitamin C (mg)	17.1–35.4 [62,63]
Vitamin K (ug)	26.7–79.2 [62,63]
Vitamin B12 (ug)	0.3–1.5 [62,63]
Vitamin B6 (mg)	0.3–0.6 [62,63]
Calcium (mg)	22.49–286.1 [58,61,62,63]
Phosphorous (mg)	37.11–450.3 [58,62,63]
Zinc (mg)	1.2–3.9 [62,63]
Potassium (mg)	323–880 [62,63]
Iron (mg)	1.8–5.3 [61,62,63]

* Not all nutrients were assessed for losses in each study; range is reported across select studies [58,61,62,63,64].

**Table 3 ijerph-18-05379-t003:** Results of studies included in Concept 2: food waste reduction interventions, nutrition, and diet quality.

Title, Author, Year	Study Type	Results
Healthy planet, healthy youth: a food systems education and promotion intervention to improve adolescent diet quality and reduce food waste [73]. Prescott et al. (2019)	Mixed-methods intervention with a nonrandomized controlled trial.	Fruit and vegetable consumption ↑ in the intervention group and ↓ in the control group. Vegetable waste was higher in the intervention group at baseline. Immediately following the intervention, there was no significant difference in salad bar vegetable waste between the intervention and control groups. At 5 months follow-up, the intervention group wasted significantly less salad bar vegetables than the control group.
Impact of a pilot school-based nutrition intervention on fruit and vegetable waste at school lunches [74]. Sharma et al. (2019)	Nonrandomized pre- and post-controlled study. Children from two schools received a “Brighter Bites” nutrition intervention while one school (control) did not receive any intervention.	Fruit and vegetable selection ↓ in the control group, but not in the intervention groups. Children in the intervention groups ↓ the amount of fruit and vegetables wasted at each meal and per item at both 8 weeks (↓ was not significant) and 16 weeks (↓ was significant) following the intervention. There was a significant ↓ in the amount of energy, carbohydrate, protein, fiber, B vitamins, and folate wasted by the intervention group.
Strategies to reduce plate waste in primary schools: experimental evaluation [75]. Martins et al. (2016)	Controlled trial. Group A: children received education on nutrition and food waste. Group B: teachers received education on food waste. Group C: control group with no intervention.	Group A ↓ soup waste in comparison to the control. This decrease was greater 1 week post intervention (−11.9%) than 3 months after the intervention (−5.8%). Group A also significantly ↓ plate waste of the main dishes 1 week post intervention (−33.9%), but this effect was no longer observed 3 months post intervention (−13.7%). Group B did not show a significant ↓ in plate waste 1 week post intervention compared with the control group. A positive effect of the intervention was evident 3 months post intervention, with a ↓in both soup waste (−5.5%) and main dish waste (−5.4%).
Effect of classroom intervention on student food selection and plate waste: evidence from a randomized control trial [76]. Serebrennikov et al. (2020)	Randomized controlled trial	The nutrition intervention had no impact on fruit and vegetable intake or food waste in the intervention group.
Sustainable and acceptable school meals through optimization analysis: an intervention study [78]. Eustachio et al. (2020)	Pre- and post-design study using an interrupted time-series analysis. Three schools participated in the study. Children received normal menus for four weeks and an optimized 4 week menu during the intervention period.	Optimization resulted in a food list that was 40% lower in GHGE while still meeting nutritional requirements. There was no significant difference in plate waste, serving waste, or consumption in any of the schools.
Successful implementation of climate-friendly, nutritious, and acceptable school meals in practice: the OPTIMAT™ intervention study [77]. Elinder et al. (2020)	Pre- and post-design study using an interrupted time series analysis. Study was conducted across 4 schools in Sweden. Children received normal menus for 4 weeks and received an optimized 4 week menu during the intervention period.	The optimized menu was 28% lower in GHGE and provided all nutrients in adequate amounts. Mean consumption and plate waste did not change significantly from baseline.
Reduced-portion entrées in a worksite and restaurant setting: impact on food consumption and waste [79]. Berkowitz et al. (2016)	Pre–post design intervention: introduction of a reduced-portion menu in two food-service operators.	The offering of reduced sized portions led to a ↓ in intakes of energy, fat, saturated fat, cholesterol, Na, fiber, calcium, potassium, and iron, and a ↓ in plate waste.

↓: decrease, ↑: increase, GHGE: greenhouse gas emissions.

**Table 4 ijerph-18-05379-t004:** Results of studies included in Concept 3: food banks/pantries and diet/nutritional quality.

Title, Author, Year	Study Type	Results
Dutch food bank parcels do not meet nutritional guidelines for a healthy diet [65]. Neter et al. (2016)	Cross-sectional study Part of the Dutch food bank study	Parcels provided excess energy, protein, and SFAs and insufficient amounts of fruit and fish. Parcels typically supplied enough fruit and fish for <2.5 days, while fiber, energy, protein, vegetables, fat, SFA, and carbohydrates were supplied for >2.5 days.
Nutritional adequacy and content of food bank parcels in Oxfordshire, UK: a comparative analysis of independent and organizational provision [81]. Fallaize et al. (2020)	Comparative analysis of Trussel Trust food bank and 9 additional independent food banks	Parcels provided excess energy, protein, carbohydrate, sugars, fat, fiber, and salt. Retinol and vitamin D were the only micronutrients for which the parcels did not meet DRVs.
Is UK emergency food nutritionally adequate? A critical evaluation of the nutritional content of UK food bank parcels [82]. Turnbull and Bhakta (2016)	Critical evaluation of the nutritional content of UK food bank parcels	Mean energy and the % energy of macronutrient intake of the emergency food parcels met the EAR and DRVs, but the constructed meal plans provided insufficient energy. A high proportion of energy supplied was from carbohydrates. Meal plans were low in fruit and vegetables and milk and dairy products in comparison to the EatWell Plate. The provision of vitamin C, calcium, magnesium, potassium, and zinc was only within LRNIs.
Nutritional quality and price of food hampers distributed by a campus food bank: a Canadian experience [83]. Jessri et al. (2014)	Time-series analysis	Hampers provided adequate energy, but insufficient animal protein and fat. All hampers did not meet requirements for vitamin A and zinc. The nutritional quality of the hampers improved significantly from 2006–2011 due to the inclusion of perishable items.
Nutritional quality of emergency foods [66]. Hoisington et al. (2011)	Cross-sectional study	66% of food supplied fell into the fruit, vegetable, grains and meat/beans and milk categories; 34% were condiments or baking supplies, discretionary calories, or combination or variety foods. Fruit and milk groups were supplied in smaller quantities than the meat/bean, grains, and vegetable groups.
The nutritional quality of food provided from food pantries: a systematic review of existing literature [70]. Simmet et al. (2017)	Systematic review (*n* = 9)	There were large variations in supply between studies. 7 studies reported that the food supply did not provide sufficient amounts/types of food for the number of days the bag was intended to last, while 2 studies reported that the food supply was adequate. Energy requirements were met or exceeded in 4 out of 6 studies that measured energy provision. Energy requirements were not met in 2 studies. In particular, dairy products and products containing vitamins A, D, and C, calcium, and zinc were lacking.
A technical and policy case study of large-scale rescue and redistribution of perishable foods by the “Leket Israel” food bank [71]. Philip et al. (2017)	Case study	The food bank functions as a wholesale operation under a business-to-business model. Food is distributed via NPOs. The food bank matches the supply of perishable foods with real-time demand so as not to redistribute waste down the supply chain. Food is obtained from an Agricultural Gleaning project, Self-Grown Farm project, and a Meal Rescue project. Dietitians are employed to cover the areas of food safety, raising awareness of nutrition and good nutritional habits, and tracking nutritional performance. In 2014, 93% of food rescued was healthy food, and 87% of food was from the fruit and vegetable groups.
The dietary quality of food pantry users: a systematic review of existing literature [69]. Simmet et al. (2017)	Systematic review (*n* = 15)	Mean energy intake, fruit and vegetable portions, and milk and dairy servings were less than recommended in all but 1 study, and mean intakes of meat and meat products were within recommendations.
Mitigating seafood waste through a bycatch donation program [72]. Watson et al. (2020)	Case study	The Prohibited Species Donation (PSD) program donates trawl fishery prohibited species catch (PSC) that would otherwise be discarded to hunger relief organizations. Over 23.5 million servings of high-quality seafood (salmon and halibut) have been redistributed to provide nutritious food to food banks, while reducing food waste.
Food-aid quality correlates positively with diet quality of food pantry users in the *Leket Israel* food bank collaborative [67]. Philip et al. (2018)	Exploratory cross-sectional study	Overall, pantry users had poor diet quality, including excessive/inadequate energy intake and micronutrient deficiencies. On average, a basket provided insufficient energy, protein, and fiber. Less than 1/3 of the baskets provided the full household requirement for most minerals and vitamins and only 1/4 of the baskets supplied the number of fruit and vegetable portions recommended per household. The food provided by Leket Israel increased the total number of healthy portions and fruit and vegetable portions and increased the fiber, vitamin, and mineral content in an average food pantry or NPO basket. Higher-quality baskets were associated with higher-quality diets, and the fruit and vegetable portions contributed by Leket Israel correlated positively with dietary quality.
Food rescue—an Australian example [84]. Lindberg et al. (2014)	Multimethod qualitative study	SecondBite provides access to fresh, nutritious food for people in need by rescuing perishable healthy food. In 2013, they rescued 3.9 million kilograms of food (almost eight million meals). They offer nutrition education and food skills programs for staff and clients and employ dietetic staff. Rescued food contributed to savings in energy, water, and CO_2_.

CO_2_: carbon dioxide, NPO: nonprofit organization, DRVs: dietary reference values, SFAs: saturated fatty acids, EAR: estimated average requirement, LRNI: lower reference nutrient intakes, RNI: recommended nutrient intake.

**Table 5 ijerph-18-05379-t005:** Key messages in literature included in Concept 4: food waste and plastic waste in nutrition or diet guidelines and policies.

Title, Author, Year	Type of Document	Key Messages	
		Food Waste	Plastic/Packaging Waste
Belgian dietary guidelines The food triangle for the Flemish community 2017 [85].	Food-based dietary guidelines	Reduce overconsumption and waste. Ecological gains can be made by reducing food waste. Recommendations: draw up a weekly menu and shopping list to reduce food waste.	N/A
Danish dietary guidelines The official dietary guidelines 2013 [86].	Food-based dietary guidelines	Avoid food waste. Recommendations: think about the food you buy and throw away, plan purchases, avoid impulse purchases, do not buy or eat more than you need, store food at the right temperature, pay attention to shelf-life, and use leftovers.	N/A
The Swiss Food Pyramid 2016 [90].	Food-based dietary guidelines	Avoiding food waste advocated as sustainable eating habit.	N/A
German dietary guidelines 2017 [89].	Food-based dietary guidelines	Food waste wastes valuable resources. Vegetables and fruit with quirks and stains also provide vitamins and minerals. Recommendations: check supplies, buy only what you need with a shopping list, and recycle/freeze leftovers. Food that is past best before date does not need to be thrown out: assess taste and smell.	Use tap instead of bottled water to save on packaging.
Qatar dietary guidelines 2015 [87].	Food-based dietary guidelines	Reduce leftovers and waste. Reduce overconsumption to avoid food waste. Recommendations: Store foods safely and properly and plan meals and shopping to decrease food waste.	Reduce overconsumption to avoid packaging waste. Cooking dried legumes instead of using canned versions reduces packaging waste. Choose foods that do not have more packaging than is required.
Canadian dietary guidelines 2019 [88].	Food-based dietary guidelines	Food waste linked to poor food skills. Wasted food has an environmental impact. Recommendations: meal planning, storing perishable foods properly, and using leftovers can help to reduce food waste.	N/A

## Data Availability

The authors confirm that the data supporting the findings of this study are available within the article.

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
