# Peer review of "Food Waste and Nutrition Quality in the Context of Public Health: A Scoping Review"

_ijerph, 2021, doi:10.3390/ijerph18105379_

Round 1

Reviewer 1 Report

The paper “Food waste and nutrition quality in the context of public health: A scoping review” is a trial to investigate the association between food waste and nutrient quality. The subject is an important issue especially from global point of view and number of tons of wasted food. Unfortunately the analysis provided has to be improved before publication in International Journal of Environmental Research and Public Health.

Major comments:

The paper has to be rewritten in the following way:

  1. All methodological description covers 80-90% of the manuscript. Such description should be added as supplementary material as well as table 1, 3, 4 and 5.
  2. There is no clear justification to limit analysis to papers published after 2010. The analysis should cover all available papers in the database (or at least from 2000) – 33 papers as a result for discussion seem too narrow to discuss such important issue
  3. Authors should synthetize results and discuss the way of food waste together with nutrition wastage. The table and short discussion are not enough. The potential reader could expect clear directions what has been done, what is current state of the art, what are the future directions. The information collected in tables seems like information taken from abstracts.
  4. The paper is a review, so the reader would expect the review of the all strategies and methodologies that were applied for food waste and nutrition quality. It would be welcome to introduce the main concepts based on separated groups as food waste and nutrition quality
  5. Instead of delivery of short abstracts in the tables, please include the extended detailed discussion regarding the thematic area covered by selected papers.

Author Response

Thank you for reviewing our manuscript and your feedback.

All methodological description covers 80-90% of the manuscript. Such description should be added as supplementary material as well as table 1, 3, 4 and 5.

Thank you for this suggestion. However, we disagree based on the following rationale:

  • The methodological description is approximately 20% of the manuscript.  We believe the information provided is essential for inclusion in the main body of the text, rather than as supplementary material, as it clearly outlines how the review was conducted and allows for reproducibility. Furthermore, the information provided in this section is in line with the PRISMA extension for scoping reviews checklist [1], which was used to structure this review, indicating all subheadings and information in this section are important for inclusion in the main text.
  • Tables 1, 3, 4, and 5 are key components of the results section. For each included source of evidence, the tables present the relevant data that were charted that relate to the review questions and objectives. This is another requirement of the PRISMA extension for scoping reviews checklist [Tricco et al., 2018] and provides the reader with information that is necessary for interpreting the remainder of the results section and discussion. As such, we believe the tables are important for inclusion in the main body of the text rather than as supplementary material. 

There is no clear justification to limit analysis to papers published after 2010. The analysis should cover all available papers in the database (or at least from 2000) – 33 papers as a result for discussion seem too narrow to discuss such important issue.

Thank you for this comment. We agree, justification was not provided in the text and this has now been amended. Please see lines 172-175. We have included the following: “The search was limited to publications in the English language published between 2010-2021. The aim of the review was to identify recent literature and determine the gaps in current research. Research in this area has increased rapidly within this timeframe, with a more urgent focus on changes needed to develop a sustainable food system.”

Analysis was limited to papers published after 2010 as we aimed to identify recent literature and to determine the gaps within current research (approximately the last 10 years). One of the core aspects of scoping reviews is that they facilitate the mapping of emerging research [Armstrong et al., 2011]. While devising the review protocol and search strategy it was decided that emerging research/literature would fall between the years 2010-2021. As this was a rapid scoping review there was also limited time for the searching and screening of articles due to deadlines for completion. It is common when undertaking a rapid review that steps are undertaken to simplify or omit certain processes in order to produce information in a timely manner [Khangura et al., 2012]. As research and interest in the area of food waste has increased in recent years (research prior to 2010 is limited), we felt it was likely that the timeframe of 2010-2021 captured the majority of relevant literature in this area while also facilitating deadlines.

Authors should synthetize results and discuss the way of food waste together with nutrition wastage. The table and short discussion are not enough. The potential reader could expect clear directions what has been done, what is current state of the art, what are the future directions. The information collected in tables seems like information taken from abstracts.

Thank you for this suggestion.  

In the results section under Concept 1 we discuss the issues of food waste and nutrient waste together briefly in the text and in Tables 1 and 2. While I agree that the link between food waste and nutrient waste is a very important issue, there were several other issues we wished to discuss within this concept (“food waste, diet quality, nutrient losses, and environmental health”), as well as several additional concepts for inclusion in the review. The overall aim of this scoping review was to map the existing literature, rather than go in-depth into one specific concept or area (e.g., food waste and nutrient waste). As such, it would be difficult to place so much emphasis on the area of food waste and nutrient waste without displacing other areas/concepts of interest or skewing the data and discussion in favor of this area. There is a wealth of information on food waste and nutrient waste and to discuss this topic to such an extent (state of the art/future directions etc.) could warrant its own separate review.

We received conflicting feedback regarding the tables. I have aimed to amend the tables and make them more concise as per another reviewer’s feedback. Please see Tables 1, 3, 4, and 5.

The paper is a review, so the reader would expect the review of the all strategies and methodologies that were applied for food waste and nutrition quality. It would be welcome to introduce the main concepts based on separated groups as food waste and nutrition quality.

Thank you for this comment. We have summarised the varying methodologies used to investigate the links between food waste and nutrition. These are summarised in lines 217-226 – “A range of publication types were included in this review, including 10 cross-sectional studies[58-67], 1 review[68], 2 systematic reviews[69,70], 2 case studies[71,72], 7 intervention studies (3 non-randomized controlled trials[73-75], 1 randomized controlled trial[76], and three pre-post-design studies[77,78]), 1 observational study[79], 1 comparative analysis[80], 1 critical evaluation[81], 1 time series[82], 1 multi-method study[83] and six food-based dietary guidelines[84-89]. Four main concepts were mapped from the resulting data (Figure 3) and the publications were grouped into these concepts as appropriate.   Characteristics of included studies and reported outcomes are summarized for each concept in Tables 1 to 5. ” As it was a scoping review, we aimed to give a brief overview of the various methodologies used rather than an in-depth discussion on this. 

We have aimed to separate our data into concepts pertaining to food waste and different aspects of nutrition and diet quality: 1) food waste and diet quality, nutrient losses, and environmental health, (2) food waste reduction interventions and diet quality, (3) food banks/pantries and diet/nutritional quality, and (4) food waste and plastic waste in nutrition or diet guidelines.

Instead of delivery of short abstracts in the tables, please include the extended detailed discussion regarding the thematic area covered by selected papers.

Thank you for this comment. We received conflicting feedback regarding the content of the tables, with another reviewer recommending that we reduce the text in the tables to allow for more comprehensive reading. We have modified the tables to reflect this feedback. Please see Tables 1, 3, 4, and 5.

References cited in responses above

Tricco, AC.; Lillie, E.; Zarin, W.; O'Brien, KK.; Colquhoun, H.; Levac, D.; Moher, D.; Peters, MD.; Horsley, T.; Weeks, L.; Hempel, S et al. PRISMA extension for scoping reviews (PRISMA-ScR): checklist and explanation. Ann Intern Med 2018, 169(7), 467-473. doi:10.7326/M18-0850.

Armstrong, R.; Hall, BJ.; Doyle, J.; Waters, E. ‘Scoping the scope’ of a cochrane review. J Public Health 2011, 33(1), 147–50.

Khangura, S.; Konnyu, K.; Cushman, R.; Grimshaw, J.; Moher, D. Evidence summaries: the evolution of a rapid review approach. Syst Rev 2012 doi:10.1186/2046-4053-1-10.

Reviewer 2 Report

The manuscript is very well prepared and authors deal with an actual issue. Authors originally connected several related issues as food loss/waste, nutrition and environmental health.

I consider this manuscript as beneficial for current research in this field. I just have several minor comments how to improve this manuscript:

In theoretical part, I miss some part dealing with households and home cooking and ways how to prevent food loss/waste (for example planning shopping behaviour, home cooking, issue of using raw ingredients or ready-made meals etc.). I recommend authors to mention at least shortly this field of consumption behaviour.

Why did you choose these three scientific database: Scopus, 139 MEDLINE, and Global Health? You should justify the reason of their selection.

In methodology, authors could describe more in-depth reasons of exclusion criteria.

In Result part, tables 1, 3, 4, 5 are interesting, but too much narrative. Especially the description in column Results (usually fourth column). I recommend to figure out somehow to improve it and make it more comprehensive for reading. Authors may create better structure, create highlights or something like this.

Last recommendation is for Discussion. Even though the COVID-19 pandemic was not the issue of the paper, authors should mention this issue, especially in connection with safety measures, potential increase of plastic packaging etc. There are already some studies dealing with this issue.

Author Response

Reviewer two comments:

In theoretical part, I miss some part dealing with households and home cooking and ways how to prevent food loss/waste (for example planning shopping behaviour, home cooking, issue of using raw ingredients or ready-made meals etc.). I recommend authors to mention at least shortly this field of consumption behaviour.

Thank you for this comment. We agree, this would add to the theoretical section. Therefore, we have included a short discussion on this in the introduction – “Food waste reduction strategies at a household level include meal planning, effective use of leftovers, correct storage of food, and the avoidance of overconsumption and excess purchasing”. Please see lines 71-73.

Why did you choose these three scientific database: Scopus, 139 MEDLINE, and Global Health? You should justify the reason of their selection.

Thank you for this suggestion. We have added justification for the use of these three databases- “Scopus and MEDLINE were selected as they would facilitate the identification of a broad range of literature and provide a comprehensive overview of the literature available in this area, which is fitting with the purpose of a scoping review. Global Health was selected as it would allow for the identification of literature in this area specifically in the context of public health, which was one of the key outcomes of this scoping review.”. Please see lines 163-167.

In methodology, authors could describe more in-depth reasons of exclusion criteria.

Thank you for this suggestion. We agree, additional reasons for exclusion should be supplied. We have justified the exclusion of literature published prior to year 2010 (this is in line with feedback received from another reviewer) – “The search was limited to literature published between 2010-2021 as the aim of this scoping review was to identify recent literature in this area and determine the gaps in current research (within the last ten years). Research in this area has also increased rapidly within this timeframe”. Please see lines 172-175.

In Result part, tables 1, 3, 4, 5 are interesting, but too much narrative. Especially the description in column Results (usually fourth column). I recommend to figure out somehow to improve it and make it more comprehensive for reading. Authors may create better structure, create highlights or something like this.

Thank you for this feedback. We have aimed to condense/restructure the information provided in each table. Please see Tables 1, 3, 4, and 5.

  • For Table 1 we retained much of the same information but have restructured the table to make it easier to read and pick out relevant information.
  • For Table 3 we removed the objectives column in order to remove large sections of text and make the table more comprehensive. We have condensed the text and used arrows to highlight increases and decreases in food waste and consumption patterns.
  • For Table 4 we retained the structure of the table but condensed the text. This has shortened the table considerably.
  • For Table 5 we retained much of the same information as the text was quite minimal here. However, we have made an effort to reduce the text slightly and have restructured the table to make it clearer which food-based dietary guidelines included messages on reducing food waste and/or plastic waste, respectively.

Last recommendation is for Discussion. Even though the COVID-19 pandemic was not the issue of the paper, authors should mention this issue, especially in connection with safety measures, potential increase of plastic packaging etc. There are already some studies dealing with this issue.

Thank you for this suggestion. We agree, this would enhance the review. I have added a section on COVID-19 and plastic waste to the discussion. “In light of the current COVID-19 pandemic it is of particular importance that the links between plastic waste, nutrition, and public health are investigated. COVID-19 had led to a change in our waste generation patterns and a surge in the demand and use of plastics, which has had subsequent effects on our waste management systems[111]. There is a perception that plastic is more hygienic which has led to a shift in consumer choice in favour of plastic packaging and single-use plastic bags[112], reusable bags have been banned in certain areas[112], lockdowns and home quarantines have led to an increase in online shopping and home delivery of food and groceries which may increase the generation of packaging waste[113], and the use of disposable utensils has increased due to convenience and safety[112]. As such, it is essential that we understand the implications that this may have for public health.”. Please see lines 458-468.

Reviewer 3 Report

This manuscript highlighted the link between food waste, nutritional losses, diet quality and environmental health. This manuscript is presented well with detailed discussion. I have following suggestion to the authors:

  • Line no. 43-45: Before the definition for food loss and food waste, author can narrate the concept of food supply chain. So, the readers can get the concept in flow.
  • Author can provide a graphical representation of the work linking the food supply, food loss and food waste.
  • Line 81-54: I presume author mean ‘rate’ as the quantity of plastics recycled or reused every year. Author can provide the integers [ton/year] to avoid confusion.
  • Line 127-130: Please rephrase this sentence. Very long and not conveying the meaning.
  • Reference section should be modified according to the MDPI guidelines. 

“Author 1, A.B.; Author 2, C.D. Title of the article. Abbreviated Journal Name YearVolume, page range.”

Author Response

Reviewer three comments:

Line no. 43-45: Before the definition for food loss and food waste, author can narrate the concept of food supply chain. So, the readers can get the concept in flow.

Thank you for this comment. We have narrated the concept of a food supply chain – “The food supply chain describes how food products pass from producer to consumer, this includes the stages of production, processing, distribution, retailing and consumption.”.  We have also integrated this into the definitions of food loss and food waste. Please see lines 44-50.

Author can provide a graphical representation of the work linking the food supply, food loss and food waste.

Thank you for this suggestion. We have included a figure linking the food supply chain, food loss, and food waste. Please see Figure 1.

Line 81-54: I presume author mean ‘rate’ as the quantity of plastics recycled or reused every year. Author can provide the integers [ton/year] to avoid confusion.

Thank you for highlighting this. We have included the percentages of plastic waste that are recycled, sent to landfill, or incinerated in order to improve upon this – “Only 30% of this plastic waste is returned for recycling, while 31% and 39% are sent to landfill or for incineration, respectively.”. Please see lines 88-91.

Line 127-130: Please rephrase this sentence. Very long and not conveying the meaning.

Thank you for this comment. We have shortened and rephrased this sentence – “Another area for inclusion was studies discussing the nutritional quality of food distributed by food redistribution systems, such as food banks and pantries. These redistribution systems have been shown to effectively redistribute food surpluses, are a key component of the food waste reduction hierarchy[22], and have the potential to impact public health.”. Please see lines 142-146.

Reference section should be modified according to the MDPI guidelines.

“Author 1, A.B.; Author 2, C.D. Title of the article. Abbreviated Journal Name Year, Volume, page range.”

Thank you for this comment. The reference section should be in line with the MDPI guidelines. Please see reference section.

Reviewer 4 Report

The authors present a scoping review of food waste and nutrition quality in the context of public health. It is well-written and timely. I appreciate the opportunity to review it. The authors have tackled a very broad topic and have managed to intelligently distill it into meaningful discussion points. I think the plastics and dietary guidelines aspects are particularly novel, and I have not seen these covered in any other food waste reviews or commentaries. My only concerns lie with the discussion section. I think there are a few areas that are either misleading or underdeveloped as noted below.

Line 304: Fruit & Vegetables are the most wasted food groups when waste is compared by weight. Other units of measure will yield other food groups (i.e. by calorie, etc.) Please specify that this is by weight.

Line 325-326: There are likely confounding factors driving this association. Fresh fruit and vegetable waste drives waste of this food group, and fresh F&V are purchased at a higher rate by higher income families. Higher income families are likely less motivated to reduce food waste due to the cheap cost of food. Given that this is just an association, I think it is misleading to isolate your explanation of the relationship without noting the high likelihood of confounding factors.

Lines 352-385: While I agree with these points, it is a bit simplistic to imply that by just increasing access to healthy foods at pantries will create a sustainable food system (lines380-354). Access is not the only issue. I see that the authors mentioned educational pantry interventions that target pantry clientele, but they don’t make the link that the problems experienced by pantry clientele are systemic issues. It will take a systemic approach to simultaneously meet the health needs of pantry clientele as well as conserve natural resources. Others have examined this topic more holistically and included relevant policies, such as https://doi.org/10.1016/j.jneb.2020.11.005.

Lines 386-393: I think this is one of the more novel aspects of the paper, and thus, it would be good to see more discussion here. What recommendations do the authors have for future dietary guidelines? Or, which country had the most robust guidelines that could be used as a starting point for nations wishing to improve their own policies? Are there findings from any of the reviewed research that could be incorporated into future dietary guidelines?

The discussion does not cover plastic waste well. It is only really discussed in the Food Based Dietary Guidelines section. This is also one of the more novel aspects of the paper that should be better addressed. What was the main takeaway about plastics that can be ascertained by your review?

Author Response

Reviewer four comments:

Line 304: Fruit & Vegetables are the most wasted food groups when waste is compared by weight. Other units of measure will yield other food groups (i.e. by calorie, etc.) Please specify that this is by weight.

Thank you for this suggestion. We have specified that this by weight – “Overall, fruit and vegetables were identified as the one of the most wasted food groups when food waste was compared by weight”. Please see lines 341-342.

Line 325-326: There are likely confounding factors driving this association. Fresh fruit and vegetable waste drives waste of this food group, and fresh F&V are purchased at a higher rate by higher income families. Higher income families are likely less motivated to reduce food waste due to the cheap cost of food. Given that this is just an association, I think it is misleading to isolate your explanation of the relationship without noting the high likelihood of confounding factors.

Thank you for this comment. This is a very valid point. We have clarified that additional factors are likely driving this association between food waste and diet quality - “However, there are likely additional factors driving this association, including household income (there may be less waste in lower income households), household size and composition (households with children are likely to waste more), and household demographics and culture [7].”. Please see lines 364-367.

Lines 352-385: While I agree with these points, it is a bit simplistic to imply that by just increasing access to healthy foods at pantries will create a sustainable food system (lines380-354). Access is not the only issue. I see that the authors mentioned educational pantry interventions that target pantry clientele, but they don’t make the link that the problems experienced by pantry clientele are systemic issues. It will take a systemic approach to simultaneously meet the health needs of pantry clientele as well as conserve natural resources. Others have examined this topic more holistically and included relevant policies, such as https://doi.org/10.1016/j.jneb.2020.11.005.

Thank you for this comment. We have adjusted the text to reflect this. “However, this is a systemic issue and improving access to nutritious food is only one step towards eradicating health inequalities and the move towards more sustainable food practices. A more long-term and holistic approach to improving dietary behaviours and increasing food security, alongside short-term solutions provided by food pantries, includes addressing systemic and social reasons for food, hunger and poverty and reducing health inequalities.” Please see lines 425-430.

Lines 386-393: I think this is one of the more novel aspects of the paper, and thus, it would be good to see more discussion here. What recommendations do the authors have for future dietary guidelines? Or, which country had the most robust guidelines that could be used as a starting point for nations wishing to improve their own policies? Are there findings from any of the reviewed research that could be incorporated into future dietary guidelines?

Thank you for your suggestion. We have incorporated your feedback into the discussion – “Future dietary guidelines should aim to integrate sustainability and healthy eating messaging, including recommendations to minimise food waste and plastic waste while improving nutrition, as this provides an opportunity to simultaneously improve public and environmental health. Overall, the Qatar FBDG were the most comprehensive in terms of sustainability and nutrition messaging. These guidelines provide healthy eating advice alongside actionable measures to reduce food and plastic waste and, as such, could be used as a starting point for countries aiming to improve and update their own food-based dietary guidelines. Going forward it may be beneficial to include more specific advice around the reduction of food waste in food-based dietary guidelines. Current messages, while important, are quite general (Figure 5). As such, it would be useful to provide more specific advice, perhaps pertaining to the most wasted food groups such as fruit and vegetables, including preparation, storage, and leftover recommendations”. Please see lines 436-448.

The discussion does not cover plastic waste well. It is only really discussed in the Food Based Dietary Guidelines section. This is also one of the more novel aspects of the paper that should be better addressed. What was the main takeaway about plastics that can be ascertained by your review?

Thank you for your comment.  We agree, the discussion on plastic waste could be improved - “Several additional areas were identified where plastic waste overlapped with public health, including microplastics in the food chain and food contact chemicals but, again, the nutrition element of the eligibility criteria was lacking. Overall, this scoping review has identified that plastic waste, nutrition, and public health are rarely discussed in combination and this highlights the need for future research, and the development of policy and practice, in this area.”. We have also tied this in with another reviewer’s suggestion to discuss plastic waste in the context of the recent COVID-19 pandemic.  Please see lines 452-468. 

Round 2

Reviewer 1 Report

This manuscript can be published in present form

Reviewer 4 Report

The authors have sufficiently addressed my comments. I have no further suggestions to offer.